# Constructing Fast Network
# through Deconstruction of Convolution

**Yunho Jeon**
School of Electrical Engineering, KAIST
jyh2986@kaist.ac.kr

**Junmo Kim**
School of Electrical Engineering, KAIST
junmo.kim@kaist.ac.kr

## Abstract

Convolutional neural networks have achieved great success in various vision tasks; however, they incur heavy resource costs. By using deeper and wider networks, network accuracy can be improved rapidly. However, in an environment with limited resources (e.g., mobile applications), heavy networks may not be usable. This study shows that naive convolution can be deconstructed into a shift operation and pointwise convolution. To cope with various convolutions, we propose a new shift operation called active shift layer (ASL) that formulates the amount of shift as a learnable function with shift parameters. This new layer can be optimized end-to-end through backpropagation and it can provide optimal shift values. Finally, we apply this layer to a light and fast network that surpasses existing state-of-the-art networks. Code is available at `https://github.com/jyh2986/Active-Shift`.

## 1 Introduction

Deep learning has been applied successfully in various fields. For example, convolutional neural networks (CNNs) have been developed and applied successfully to a wide variety of vision tasks. In this light, the current study examines various network structures[15, 19, 21, 20, 22, 6, 7, 9]. In particular, networks are being made deeper and wider because doing so improves accuracy. This approach has been facilitated by hardware developments such as graphics processing units (GPUs).

However, this approach increases the inference and training times and consumes more memory. Therefore, a large network might not be implementable in environments with limited resources, such as mobile applications. In fact, accuracy may need to be sacrificed in such environments. Two main types of approaches have been proposed to avoid this problem. The first approach is network reduction via pruning[5, 4, 17], in which the learned network is reduced while maximizing accuracy. This method can be applied to most general network architectures. However, it requires additional processes after or during training, and therefore, it may further increase the overall amount of time required for preparing the final networks.

The second approach is to use lightweight network architectures[10, 8, 18] or new components[13, 23, 26] to accommodate limited environments. This approach does not focus only on limited resource environments; it can provide better solutions for many applications by reducing resource usage while maintaining or even improving accuracy. Recently, grouped or depthwise convolution has attracted attention because it reduces the computational complexity greatly while maintaining accuracy. Therefore, it has been adopted in many new architectures[2, 24, 8, 18, 26].

Decomposing a convolution is effective for reducing the number of parameters and computational complexity. Initially, a large convolution was decomposed using several small convolutions[19, 22]. Binary pattern networks[13] have also been proposed to reduce network size. This raises the

question of what the atomic unit is for composing a convolution. We show that a convolution can be deconstructed into two components: 1×1 convolution and shift operation.

Recently, shift operations have been used to replace spatial convolutions[23] and reduce the number of parameters and computational complexity. In this approach, shift amounts are assigned heuristically by grouping input channels. In this study, we formulate the shift operation as a learnable function with shift parameters and optimize it through training. This generalization affords many benefits. First, we do not need to assign shift values heuristically; instead, they can be trained from random initializations. Second, we can simulate convolutions with large receptive fields such as a dilated convolution[1, 25]. Finally, we can obtain a significantly improved tradeoff between performance and complexity.

The contributions of this paper are summarized as follows:

- We deconstruct a heavy convolution into two atomic operations: 1×1 convolution and shift operation. This deconstruction can greatly reduce the parameters and computational complexity.
- We propose an active shift layer (ASL) with learnable shift parameters that allows optimization through backpropagation. It can replace the existing spatial convolutions while reducing the computational complexity and inference time.
- The proposed method is used to construct a light and fast network. This network shows state-of-the-art results with fewer parameters and low inference time.

## 2   Related Work

**Decomposition of Convolution** VGG[19] decomposes a 5×5 convolution into two 3×3 convolutions to reduce the number of parameters and simplify network architectures. GoogleNet[22] uses 1×7 and 7×1 spatial convolutions to simulate a 7×7 convolution. Lebedev et al. [16] decomposed a convolution with a sum of four convolutions with small kernels. Recently, depthwise separable convolution has been shown to achieve similar accuracy to naive convolution while reducing computational complexity[2, 8, 18].

In addition to decomposing a convolution into other types of convolutions, a new unit has been proposed. A fixed binary pattern[13] has been shown to be an efficient alternative for spatial convolution. A shift operation[23] can approximately simulate spatial convolution without any parameters and floating point operations (FLOPs). An active convolution unit (ACU)[11] and deformable convolution[3] showed that the input position of convolution can be learned by introducing continuous displacement parameters.

**Mobile Architectures** Various network architectures have been proposed for mobile applications with limited resources. SqueezeNet[10] designed a fire module for reducing the number of parameters and compressed the trained network to a very small size. ShuffleNet[26] used grouped 1×1 convolution to reduce dense connections while retaining the network width and suggested a shuffle layer to mix grouped features. MobileNet[18, 8] used depthwise convolution to reduce the computational complexity.

**Network Pruning** Network pruning[5, 4, 17] is not closely related to our work. However, this methodology has similar aims. It reduces the computational complexity of trained architectures while maintaining the accuracy of the original networks. It can be also applied to our networks for further optimization.

## 3   Method

The basic convolution has many weight parameters and large computational complexity. If the dimension of the weight parameter is $D \times C \times K$, the computation complexity (in FLOPs) is

$$(D \times C \times K) \times (W \times H), \tag{1}$$

where $K$ is the spatial dimension of the kernel (e.g., $K$ is nine for 3×3 convolution); $C$ and $D$ are the numbers of input and output channels, respectively; and the spatial dimension of the input feature is $W \times H$.

## 3.1 Deconstruction of Convolution

The basic convolution for one spatial position can be formulated as follows:

$$\tilde{y}_{d,m,n} = \sum_c \sum_k \tilde{w}_{d,c,k} \cdot \tilde{x}_{c,m+i_k,n+j_k} = \sum_k \sum_c \tilde{w}_{d,c,k} \cdot \tilde{x}_{c,m+i_k,n+j_k}, \tag{2}$$

where $\tilde{x}_{\cdot,\cdot,\cdot}$ is an element of the $C \times W \times H$ input tensor, and $\tilde{y}_{\cdot,\cdot,\cdot}$ is an element of the $D \times W \times H$ output tensor. $w_{\cdot,\cdot,\cdot}$ is an element of the $D \times C \times K$ weight tensor $\tilde{W}$. $k$ is the spatial index of the kernel, which points from top-left to bottom-right of a spatial dimension of the kernel. $i_k$ and $j_k$ are displacement values for the corresponding kernel index $k$. The above equation can be converted to a matrix multiplication:

$$
\boldsymbol{Y} = \boldsymbol{W}^+ \times \boldsymbol{X}^+ = \left[ \tilde{\boldsymbol{W}}_{:,:,1} \ \tilde{\boldsymbol{W}}_{:,:,2} \ ... \ \tilde{\boldsymbol{W}}_{:,:,K} \right] \times \begin{bmatrix} \boldsymbol{X}^1_{:,:} \\ \boldsymbol{X}^2_{:,:} \\ ... \\ \boldsymbol{X}^K_{:,:} \end{bmatrix}
$$

$$
= \sum_k \tilde{\boldsymbol{W}}_{:,:,k} \times \boldsymbol{X}^k_{:,:} = \sum_k \tilde{\boldsymbol{W}}_{:,:,k} \times S_k(\boldsymbol{X}) \tag{3}
$$

where $\boldsymbol{W}^+$ and $\boldsymbol{X}^+$ are reordered matrices for the weight and input, respectively. $\tilde{\boldsymbol{W}}_{:,:,k}$ represents the $D \times C$ matrix corresponding to the kernel index $k$. $\boldsymbol{X}$ is a $C \times (W \cdot H)$ matrix, and $\boldsymbol{X}^k_{:,:}$ is a $C \times (W \cdot H)$ matrix that represents a spatially shifted version of input matrix $\boldsymbol{X}$ with shift amount $(i_k, j_k)$. Then, $\boldsymbol{W}^+$ becomes a $D \times (K \cdot C)$ matrix, and $\boldsymbol{X}^+$ becomes a $(K \cdot C) \times (W \cdot H)$ matrix. The output $\boldsymbol{Y}$ forms a $D \times (W \cdot H)$ matrix.

Eq. (3) shows that the basic convolution is simply the sum of $1 \times 1$ convolutions on shifted inputs. The shifted input $\boldsymbol{X}^k_{:,:}$ can be formulated using the shift function $S_k$ that maps the original input to the shifted input corresponding to the kernel index $k$. The conventional convolution uses the usual shifted input with integer-valued shift amounts for each kernel index $k$. As an extreme case, if we can share the shifted inputs regardless of the kernel index, that is, $S_k(\boldsymbol{X}) = S(\boldsymbol{X})$, this simplifies to just one pointwise (i.e., $1 \times 1$) convolution and greatly reduces the computation complexity:

$$\sum_k \tilde{\boldsymbol{W}}_{:,:,k} \times S_k(\boldsymbol{X}) = \sum_k \tilde{\boldsymbol{W}}_{:,:,k} \times S(\boldsymbol{X}) = (\sum_k \tilde{\boldsymbol{W}}_{:,:,k}) \times S(\boldsymbol{X}) = \boldsymbol{W} \times S(\boldsymbol{X}) \tag{4}$$

However, as in this extreme case, if only one shift is applied to all input channels, the receptive field of convolution is too limited, and the network will provide poor results. To overcome this problem, ShiftNet[23] introduced grouped shift that applies different shift values by grouping input channels. This shift function can be represented by Eq. (5) and is followed by the single pointwise convolution $\boldsymbol{Y} = \boldsymbol{W} \times S_G(\boldsymbol{X})$.

$$
S_G(\boldsymbol{X}) = \begin{bmatrix} \boldsymbol{X}^1_{1:n,:} \\ \boldsymbol{X}^2_{n+1:2n,:} \\ ... \\ \boldsymbol{X}^G_{(G-1)n+1:,:} \end{bmatrix} \tag{5}
\qquad
S_C(\boldsymbol{X}) = \begin{bmatrix} \boldsymbol{X}^1_{1,:} \\ \boldsymbol{X}^2_{2,:} \\ ... \\ \boldsymbol{X}^C_{C,:} \end{bmatrix} \tag{6}
$$

where $n$ is the number of channels per kernel, and it is the same as $\lfloor C/K \rfloor$. $G$ is the number of shift groups. If $C$ is a multiple of $K$, $G$ is the same as $K$, otherwise, $G$ is $K+1$. The shift function applies different shift values according to the group number and not by the kernel index. The amount of shift is assigned heuristically to cover all kernel dimensions, and pointwise convolution is applied before and after the shift layer to make it invariant to the permutation of input and output channels.

## 3.2 Active Shift Layer

Applying the shift by group is a little artificial, and if the kernel size is large, the number of input channels per shift is reduced. To solve this problem, we suggest a depthwise shift layer that applies

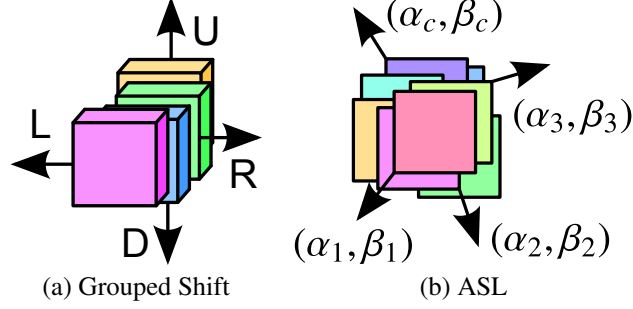

(a) Grouped Shift      (b) ASL

Figure 1: Comparison of shift operation: (a) shifting is applied to each group and the shift amount is assigned heuristically[23] and (b) shifting is applied to each channel using shift parameters and they are optimized by training.

different shift values for each channel (Eq. (6)). This is similar to decomposing a convolution into depthwise separable convolutions. Depthwise separable convolution first makes features with depthwise convolution and mixes them with $1\times1$ convolutions. Similarly, a depthwise shift layer shifts each input channel, and shifted inputs are mixed with $1\times1$ convolutions.

Because the shifted input $S_C(\boldsymbol{X})$ goes through single $1\times1$ convolutions, this reduces the computational complexity by a factor of the kernel size $K$. More importantly, by removing both the spatial convolution and sparse memory access, it is possible to construct a network with only dense operations like $1\times1$ convolutions. This can provide a greater speed improvement than that achieved by reducing only the number of FLOPs.

Next, we consider how to assign the shift value for each channel. The exhaustive search over all possible combinations of assigning the shift values for each channel is intractable, and assigning values heuristically is suboptimal. We formulated the shift values as a learnable function with the additional shift parameter $\theta_s$ that defines the amount of shift of each channel (Eq. (7)). We called the new component the active shift layer (ASL).

$$\theta_s = \{(\alpha_c, \beta_c)|1 \le c \le C\} \tag{7}$$

where $c$ is the index of the channel, and the parameters $\alpha_c$ and $\beta_c$ define the horizontal and vertical amount of shift, respectively. If the parameter is an integer, it is not differentiable and cannot be optimized. We can relax the integer constraint and allow $\alpha_c$ and $\beta_c$ to take real numbers, and the value for non-integer shift can be calculated through interpolation. We used bilinear interpolation following [11]:

$$\begin{aligned}\tilde{x}_{c,m+\alpha_c,n+\beta_c} =& Z_c^1 \cdot (1 - \Delta\alpha_c) \cdot (1 - \Delta\beta_c) + Z_c^3 \cdot \Delta\alpha_c \cdot (1 - \Delta\beta_c) \\ &+ Z_c^2 \cdot (1 - \Delta\alpha_c) \cdot \Delta\beta_c + Z_c^4 \cdot \Delta\alpha_c \cdot \Delta\beta_c,\end{aligned} \tag{8}$$

$$\Delta\alpha_c = \alpha_c - \lfloor\alpha_c\rfloor, \Delta\beta_c = \beta_c - \lfloor\beta_c\rfloor, \tag{9}$$

where $(m, n)$ is the spatial position of the feature map, and $Z_c^i$ are the four nearest integer points for bilinear interpolation:

$$\begin{aligned}Z_c^1 &= x_{c,m+\lfloor\alpha_c\rfloor,n+\lfloor\beta_c\rfloor}, Z_c^2 = x_{c,m+\lfloor\alpha_c\rfloor,n+\lfloor\beta_c\rfloor+1}, \\ Z_c^3 &= x_{c,m+\lfloor\alpha_c\rfloor+1,n+\lfloor\beta_c\rfloor}, Z_c^4 = x_{c,m+\lfloor\alpha_c\rfloor+1,n+\lfloor\beta_c\rfloor+1}.\end{aligned} \tag{10}$$

By using interpolations, the shift parameters are differentiable, and therefore, they can be trained through backpropagation. With the shift parameter $\theta_s$, a conventional convolution can be formulated as follows with ASL $S_C^{\theta_s}$:

$$\boldsymbol{Y} = \boldsymbol{W} \times S_C^{\theta_s}(\boldsymbol{X}) = \boldsymbol{W} \times \begin{bmatrix} \boldsymbol{X}_{1,:}^{(\alpha_1, \beta_1)} \\ \boldsymbol{X}_{2,:}^{(\alpha_2, \beta_2)} \\ ... \\ \boldsymbol{X}_{C,:}^{(\alpha_C, \beta_C)} \end{bmatrix} \tag{11}$$

| Layer | Inference time | FLOPs |
|---|---|---|
| 3×3 dw-conv | 39 ms | 29M |
| 1×1 conv | 11 ms | 206M |
| BN+Scale/Bias[a] | 3 ms | <5M |
| ReLU | 5 ms | <5M |
| Eltwise Sum | 2 ms | <5M |

Table 1: Comparison of inference time vs. number of FLOPs. A smaller number of FLOPs does not guarantee fast inference.

[a]The implementation of BN of Caffe [12] is not efficient because it is split with two components. We integrated them for fast inference.

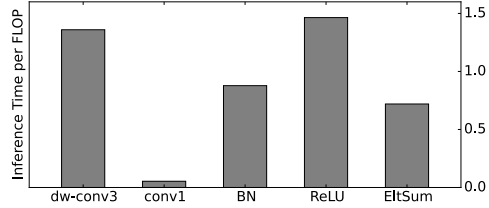

Figure 2: Ratio of time to FLOPs. This represents the inference time per 1M FLOPs. A lower value means that the unit runs efficiently. Time is measured using an Intel i7-5930K CPU with a single thread and averaged over 100 repetitions.

ASL affords many advantages compared to the previous shift layer[23] (Fig. 1). First, shift values do not need to be assigned manually; instead, they are learned through backpropagation during training. Second, ASL does not depend on the original kernel size. In previous studies, if the kernel size changed, the group for the shift also changed. However, ASL is independent of kernel size, and it can enlarge its receptive field by increasing the amount of shift. This means that ASL can mimic dilated convolution[1, 25]. Furthermore, there is no need to permutate to achieve the invariance properties in ASL; therefore, it is not necessary to use 1×1 convolution before ASL. Although our component has parameters unlike the heuristic shift layer[23], these are almost negligible compared to the number of convolution parameters. If the input channel is $C$, ASL has only $2 \cdot C$ parameters.

### 3.3 Trap of FLOPs

FLOPs is widely used for comparing model complexity, and it is considered proportional to the run time. However, a small number of FLOPs does not guarantee fast execution speed. Memory access time can be a more dominant factor in real implementations. Because I/O devices usually access memory in units of blocks, many densely packed values might be read faster than a few numbers of largely distributed values. Therefore, the implementability of an efficient algorithm in terms of both FLOPs and memory access time would be more important. Although a 1×1 convolution has many FLOPs, this is a dense matrix multiplication that is highly optimized through general matrix multiply (GEMM) functions. Although depthwise convolution reduces the number of parameters and FLOPs greatly, this operation needs fragmented memory access that is not easy to optimize.

For gaining a better understanding, we performed simple experiments to observe the inference time of each unit. These experiments were conducted using a 224×224 image with 64 channels, and the output channel dimension of the convolutions is also 64. Table 1 shows the experimental results. Although 1×1 convolution has a much larger number of FLOPs than 3×3 depthwise convolution, the inference of 1×1 convolution is faster than that of 3×3 depthwise convolution.

The ratio of inference time to FLOPs can be a useful measure for analyzing the efficiency of layers (Fig. 2). As expected, 1×1 convolution is most efficient, and the other units have similar efficiency. Here, it should be noted that although axillary layers are fast compared to convolutions, they are not that efficient from an implementation viewpoint. Therefore, to make a fast network, we have to also consider these auxiliary layers. These results are derived using the popular deep learning package Caffe [12], and it does not guarantee an optimized implementation. However, it shows that we should not rely only on the number of FLOPs alone to compare the network speed.

## 4 Experiment

To demonstrate the performance of our proposed method, we conducted several experiments with classification benchmark datasets. For ASL, the shift parameters are randomly initialized with uniform distribution between -1 and 1. We used a normalized gradient following ACU[11] with an initial learning rate of 1e-2. Input images are normalized for all experiments.

## 4.1 Experiment on CIFAR-10/100

We conducted experiments to verify the basic performance of ASL with the CIFAR-10/100 dataset [14] that contains 50k training and 10k test 32×32 images. We used conventional pre-processing methods [7] to pad four pixels of zeros on each side, flipped horizontally, and cropped randomly. We trained 64k iterations with an initial learning rate of 0.1 and multiplied by 0.1 after 32k and 48k iterations.

We compared the results with those of ShiftNet[23], which applied shift operations heuristically. The basic building block is BN-ReLU-1×1 Conv-BN-ReLU-ASL-1×1 Conv order. The network size is controlled by multiplying the expansion rate ($\varepsilon$) on the first convolution of each residual block. Table 3 shows the results; ours are consistently better than the previous results by a large margin. We found that widening the base width is more efficient than increasing the expansion rate; this increases the width of all layers in a network. With the same depth of 20, the network with base width of 46 achieved better accuracy than that with a base width of 16 and expansion rate of 9. The last row of the table shows that our method provided better results with fewer parameters and smaller depth. Because increasing depth caused an increase in inference time, this approach is also better in terms of inference speed.

Interestingly, our proposed layer not only reduces the computational complexity but also could improve the network performance. Table 2 shows a comparison result with the network using depthwise convolution. A focus on optimizing resources might reduce the accuracy; nonetheless, our proposed architecture provided better results. This shows the possibility of extending ASL to a general network structure. A network with ASL runs faster in terms of inference time, and training with ASL is also much faster owing to the reduction in sparse memory access and the number of BNs.

Table 2: Comparison with networks for depthwise convolution. ASL makes the network faster and provides better results. B and DW denote the BN-ReLU layer and depthwise convolution, respectively. For a fair comparison of BN, we also conducted experiments on a network without BN-ReLU between depthwise convolution and last 1×1 convolution(1B-DW3-1).

| Building block | C10 | Inference Time (CPU[a]) | Training Time (GPU[b]) |
|---|---|---|---|
| 1B-DW3-B-1 | 94.16 | 16 ms | 9h03 |
| 1B-DW3-1 | 93.97 | 15 ms | 7h41 |
| 1B-ASL-1(ours) | **94.5** | **10.6 ms** | **5h53** |

[a]Intel i7-5930K
[b]GTX Titan X(Maxwell)

Fig. 3 shows an example of the shift parameters of each layer after optimizations (ASNet with base width of 88). Large shift parameter values mean that a network can view a large receptive field. This is similar to the cases of ACU[11] and dilated convolution[1, 25], both of which enlarge the receptive field without additional weight parameters. An interesting phenomenon is observed; whereas the shift values of the other layers are irregular, the shift values in the layer with stride 2 (stage2/shift1, stage3/shift1) can be seen to tend to the center between the pixels. This seems to compensate for reducing the resolution. These features make ASL more powerful than conventional convolutions, and it could result in higher accuracy.

## 4.2 Experiment on ImageNet

To prove the generality of the proposed method, we conducted experiments with an ImageNet 2012 classification task. We did not apply intensive image augmentation; we simply used a randomly flipped image with 224×224 cropped from 256×256 following Krizhevsky et al. [15]. The initial learning rate was 0.1 with a linearly decaying learning rate, and the weight decay was 1e-4. We trained 90 epochs with 256 batch size.

Our network is similar to a residual network with bottleneck blocks[6, 7], and Table 6 shows our network architecture. All building blocks are in a residual path, and we used a pre-activation residual[7]. We used the same basic blocks as in the previous experiment, and we used only one spatial convolution for the first layer. Because increasing the depth also increases the number of auxiliary

Table 3: Comparison with ShiftNet. Our results are better by a large margin. The last row shows that our result is better with a smaller number of parameters and depth.

| Depth[1] | Base Width | $\varepsilon$ | Param(M) | ShiftNet[23] C10 | ShiftNet[23] C100 | ASNet(ours) C10 | ASNet(ours) C100 |
|---|---|---|---|---|---|---|---|
| 20 | 16 | 1 | 0.035 | 86.66 | 55.62 | 89.14 | 63.43 |
| 20 | 16 | 3 | 0.1 | 90.08 | 62.32 | 91.62 | 68.83 |
| 20 | 16 | 6 | 0.19 | 90.59 | 68.64 | 92.54 | 70.68 |
| 20 | 16 | 9 | 0.28 | 91.69 | 69.82 | 92.93 | 71.83 |
| 20 | 46 | 1 | 0.28 | - | - | 93.52 | 73.07 |
| 110 | 16 | 6 | 1.2 | 93.17 | 72.56 | 93.73 | 73.46 |
| 20 | 88 | 1 | 0.99 | - | - | 94.53 | 76.73 |

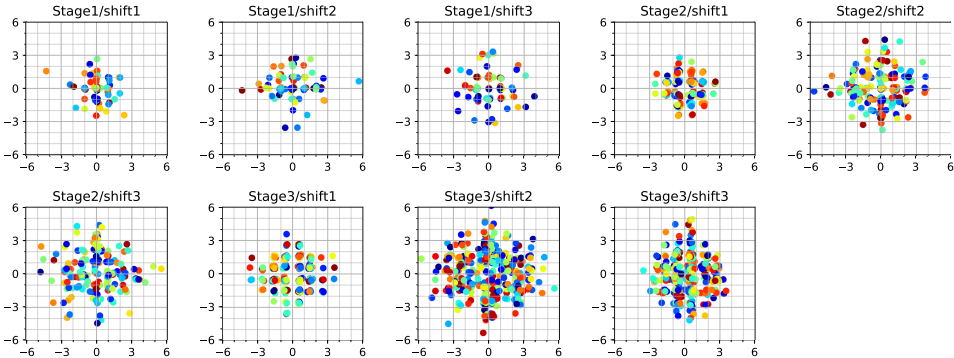

Figure 3: Trained shift values of each layer. Shifted values are scattered to various positions. This enables the network to cover multiple receptive fields.

layers, we expanded the network width to increase the accuracy as in previous studies[8, 18, 23, 26]. The network width is controlled by the base width $w$.

Table 5 shows the result for ImageNet, and our method obtains much better results with fewer parameters compared to other networks. In particular, we surpass ShiftNet by a large margin; this indicates that ASL is much more powerful than heuristically assigned shift operations. In terms of inference time, we compared our network with MobileNetV2, one of the best-performing networks with fast inference time. MobileNetV2 has fewer FLOPs compared to our network; however, this does not guarantee a fast inference time, as we noted in section 3.3. Our network runs faster than MobileNetV2 because we have smaller depth and no depthwise convolutions that run slow owing to memory access (Fig. 4).

### 4.3 Ablation Study

Although both ShiftNet[23] and ASL use shift operation, ASL achieved much better accuracy. This is because of a key characteristic of ASL, namely, that it learns the real-valued shift amounts using the network itself. To clarify the factor of improvement according to the usage of ASL, we conducted additional experiments using ImageNet (AS-ResNet-w32). Table 4 shows the top-1 accuracy with the amount of improvement inside parenthesis. *Grouped Shift* (GS) indicates the same method as ShiftNet. *Sampled Real* (SR) indicates cases in which the initialization of the shift values was sampled from a Gaussian distribution with standard deviation 1 to imitate the final state of shift values trained by ASL. Similarly, the values for *Sampled Integer* (SI) are obtained from a Gaussian distribution, but rounds a sample of real numbers to an integer point. *Training Real* (TR) is the same as our proposed method, and only TR learns the shift values.

Comparing GS with SI suggests that random integer sampling is slightly better than heuristic assignment owing to the potential expansion of the receptive field as the amount of shift can be larger than 1. The result of SR shows that a relaxation of the shift values to the real domain, another key

characteristic of ASL, turned out to be even more helpful. In terms of learning shift values, TR achieved the largest improvement, which was around two times that achieved by SR. These results show the effectiveness of learning real-valued shift parameters.

Table 4: Ablation study using AS-ResNet-w32 on ImageNet. The improvement by using ASL originated from expanding the domain of a shift parameter from integer to real and learning shifts.

| Method | Shifting Domain | Initialization | Learning Shift | Top-1 |
|---|---|---|---|---|
| Grouped Shift | Integer | Heuristic | X | 59.8 (-) |
| Sampled Integer | Integer | $\mathcal{N}(0,1)$ | X | 60.1 (+0.3) |
| Sampled Real | Real | $\mathcal{N}(0,1)$ | X | 61.9 (+2.1) |
| Training Real | Real | $\mathcal{U}[-1,1]$ | O | 64.1 (+4.3) |

Table 5: Comparison with other networks. Our networks achieved better results with similar number of parameters. Compared to MobileNetV2, our network runs faster although it has a larger number of FLOPs. Table is sorted by descending order of the number of parameters.

| Network | Top-1 | Top-5 | Param(M) | FLOPs(M) | Inference Time[a] CPU(ms) | GPU(ms) |
|---|---|---|---|---|---|---|
| MobileNetV1[8] | 70.6 | - | 4.2 | 569 | - | - |
| ShiftNet-A[23] | 70.1 | 89.7 | 4.1 | 1.4G | 74.1 | 10.04 |
| MobileNetV2[18] | 71.8 | **91** | 3.47 | 300 | 54.7 | 7.07 |
| AS-ResNet-w68(ours) | **72.2** | 90.7 | 3.42 | 729 | 47.9 | 6.73 |
| ShuffleNet-×1.5[26] | 71.3 | - | 3.4 | 292 | - | - |
| MobileNetV2-×0.75 | 69.8 | **89.6** | 2.61 | 209 | 40.4 | 6.23 |
| AS-ResNet-w50(ours) | **69.9** | 89.3 | 1.96 | 404 | 32.1 | 6.14 |
| MobileNetV2-×0.5 | 65.4 | 86.4 | 1.95 | 97 | 26.8 | 5.73 |
| MobileNetV1-×0.5 | 63.7 | - | 1.3 | 149 | - | - |
| SqueezeNet[10] | 57.5 | 80.3 | 1.2 | | - | - |
| ShiftNet-B | 61.2 | 83.6 | 1.1 | 371 | 31.8 | 7.88 |
| AS-ResNet-w32(ours) | **64.1** | **85.4** | 0.9 | 171 | 18.7 | 5.37 |
| ShiftNet-C | 58.8 | 82 | 0.78 | | - | - |

[a]Measured by Caffe [12] using an Intel i7-5930K CPU with a single thread and GTX Titan X (Maxwell). Inference time for MobileNet and ShiftNet (including FLOPs) are measured by using their network description.

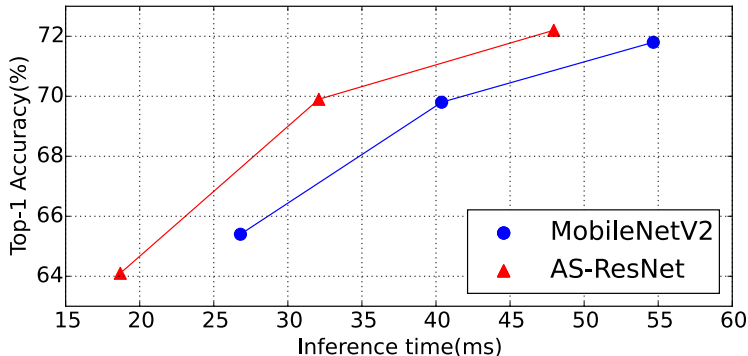

Figure 4: Comparison with MobileNetV2[18]. Our network achieves better results with the same inference time.

Table 6: Network structure for AS-ResNet. Network width is controlled by base width $w$

| Input size | Output size | Operator | Output channel | Repeat | stride |
|---|---|---|---|---|---|
| $224^2$ | $112^2$ | $3{\times}3$ conv | $w$ | 1 | 2 |
| $112^2$ | $112^2$ | basic block | $w$ | 1 | 1 |
| $112^2$ | $56^2$ | basic block | $w$ | 3 | 2 |
| $56^2$ | $28^2$ | basic block | $2w$ | 4 | 2 |
| $28^2$ | $14^2$ | basic block | $4w$ | 6 | 2 |
| $14^2$ | $7^2$ | basic block | $8w$ | 3 | 2 |
| $7^2$ | 1 | global avg-pool | - | 1 | - |
| 1 | 1 | fc | 1000 | 1 | - |

## 5  Conclusion

In this study, we deconstruct convolution to a shift operation followed by pointwise convolution. We formulate a shift operation as a function having additional parameters. The amount of shift can be learned end-to-end through backpropagation. The ability of learning shift values can help mimic various type of convolutions. By sharing the shifted input, the number of parameters and computational complexity can be reduced greatly. We also showed that using ASL could improve the network accuracy while reducing the network parameters and inference time. By using the proposed layer, we suggested a fast and light network and achieved better results compared to those of existing networks. The use of ASL for more general network architectures could be an interesting extension of the present study.

## Footnotes

[1]Depths are counted without shift layer, as noted in a previous study

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
