[Reviews · NeurIPS 2018]

Reviewer 1



I think the authors provided a solid rebuttal. My concerns with a) comparison with ShiftNet, b) sampling vs learning shift and c) comparing also GPU times were all sufficiently addressed. I am thus recommending to accept the paper. =============== The paper proposes a layer coined as ASL that combines learnable shifts with 1x1 convolutions. The overall framework, which can be viewed as a generalization of ShiftNet[23]. The ASL layer can fully capture the normal convolution (given enough channels) and can automatically trade off kernel size and the number of (traditional) channels, having only one hyperparameter: the number of output channels. For inference, the ASL only requires 1x1 convolution, and shifting operations (with interpolation for fractional pixel shifts). The authors do not mention the cost of the interpolation, but assuming that is small, the overall complexity is little compared to plain 1x1 convolution. The paper is overall well written and easy to follow, and I think the proposed approach is sound and elegant. However I'm not an expert in this field and thus have a harder time assessing the significance of the experimental results. Overall, the network seems to be faster than competing approaches, but I have some concerns: * When comparing ASL to ShiftNet, why is not the exact same architecture used and only heuristic shifts vs learned shifts compared * It is a bit surprising that all the learned shifts look like gaussians in the range -3...3 in Fig 3. What would happen if you just randomly sampled the shifts instead? * How do the timings change when testing on a GPU instead of CPU?

Reviewer 2



This work suggests a new creative and efficient way of factorizing full convolutions into 1x1 convolution and parametrizable shift operations for convolutional computer vision models. While this method is similar to depth-wise convolution it gives a new more efficient parametrization that ends up with cheaper convolution models (less parameters), less overfitting and faster training. The approach is validated experimentally on the CIFAR-10/100 and ImageNet benchmark and is compared with low-resource models like ShuffleNet and MobileNet. This is certainly a very creative work with a non-obvious idea that leads to considerable improvements of various aspects of the training and evaluation resource usage while improving generalization at the same time. Low resource computer vision is an increasingly important area and this work provides convincing evidence for the efficiency of this creative methodology.

Reviewer 3



In this paper, the authors propose a network compression framework by decomposing conventional convolution operator into a depth-wise learnable shifting operation, followed by 1x1 convolution. Intuitively, this work can be approximately seen as an improvement over MobileNet by replacing the depth-wise convolution into shift operations from [23]/[11]. The idea of learning continuous displacement in shift kernels also shares certain similar spirit to deformable convolution. Pros of the paper: The addressed problem is clearly of importance to the machine learning community and practical applications. The proposed work is technically correct and is clearly presented. The whole paper is easy to follow. The presented method achieved promising, state of the art results in the experiments. Cons of the paper: The novelty of this paper is somewhat weakened given the existence of previous related-works (ShiftNet/Active Convolution/MobileNet). In particular, the proposed work can be regarded as a combination of 1x1 convolution (can be found in MobileNet) and shift operations (similar ideas can be found in ShiftNet/Active Convolution). However, the authors provided some insights on the deconstruction of convolution in Section 3.1, which makes sense and justified the intuition of this combination. A concern is that, since FLOP may not be an accurate indicator of the inference speed, the final speed performance can depend largely on the practical network architecture/implementation which is case-by-case. The fact that the network architectures are not unified between different comparing methods leaves the space for one to empirically pick a configuration that favors the proposed compression method. An example is what the authors mentioned in lines 188-189: "We noted that increasing depth caused an increase in inference time; therefore, a shallower network is preferred." As a result, it is not easy to find whether the inference speed gain mostly comes from the selected implementation/architecture or from the proposed method itself. (Of course, the same problem applies to previous methods as well.) ======================================================================================== Update: I have read both the rebuttal and the other reviews. I decide to increase the score given the authors' additional results in response to R1. However, although the authors responded to my questions in the rebuttal, the answer somewhat dodged my concern without directly addressing it. In particular, the additional results from DW-ResNet proves that the proposed ASL module is better than MobileNet's depth-wise conv + 1x1 conv under current architecture. Since the authors' have specifically chosen the network architecture to be shallow and wide, whether the inference speed gain from the proposed ASL module is largely conditioned on the choice network architectures remains a question. It will be better if the authors can additionally include results on the comparison with ShiftNet/MobileNet under their proposed architectures (Say ASNet-110 vs ShiftNet-110). This will help people better understand the limits or properties of the proposed ASL module.